# The Effects of Agaro-Oligosaccharides Produced by Marine Bacteria (*Rheinheimera* sp. (HY)) Possessing Agarose-Degrading Enzymes on Myotube Function

**DOI:** 10.3390/md22110515

**Published:** 2024-11-14

**Authors:** Youshi Huang, Takuya Hirose, Jyh-Ming Tsai, Katsuya Hirasaka

**Affiliations:** 1Graduate School of Fisheries and Environmental Sciences, Nagasaki University, Nagasaki 8528521, Japan; huangyoushi@gmail.com (Y.H.);; 2Department of Marine Biotechnology, National Kaohsiung University of Science and Technology, Kaohsiung 811213, Taiwan; 3Organization for Marine Science and Technology, Nagasaki University, Nagasaki 8528521, Japan

**Keywords:** agarase, agaro-oligosaccharides, C2C12, muscle

## Abstract

Agarase and its metabolites are reported to have applications in a variety of fields, but there have been few studies of the effects of agaro-oligosaccharide hydrolysate on muscle function. In this study, we analyzed the functionality of agarase and its metabolites in bacteria isolated from seawater. A bacterium with agar-degrading activity was isolated from Shimabara, Nagasaki, Japan. Through 16S rRNA sequence alignment, it was identified as being closely related to *Rheinheimera* sp. WMF-1 and was provisionally named *Rheinheimera* sp. (HY). Crude enzymes derived from this bacterium demonstrated an ability to hydrolyze various polysaccharides, including agar, agarose, and starch, with the highest specificity observed for agarose. The optimum pH and temperature were pH 10 and 50 °C. A glycoside bond specificity analysis of enzymatic activity indicated the cleavage of the α-linkage. Next, we investigated the functional effects of agaro-oligosaccharides on C2C12 myotubes. Treatment with 10–30 kDa oligosaccharides significantly increased the hypertrophy rate, diameter, and expression of myosin heavy-chain genes in C2C12 myotubes. These results indicate that the agaro-oligosaccharides produced by the enzymes identified in this study improve muscle mass, suggesting their potential contribution to muscle function.

## 1. Introduction

Polysaccharides are derived from plants, microbes, animals, and algae. Alginate, a polysaccharide of high molecular weight, has been isolated from red and brown algae. The main types of polysaccharides include agar, guar, keratin, alginate, carrageenan, gum arabic, starch, and pectin [1]. Agar has been used as a food source in Japan for several centuries. During the seventeenth and eighteenth centuries, its use spread from Japan to other East Asian countries [2]. It is also known as agar-agar, and is a polysaccharide obtained from the cell walls of red algae (Rhodophyta). It is composed of approximately 70% agarose (also referred to as neutral agar), which contains approximately 2% sulfate groups, and 30% agaropectin (also known as acidic agar), which contains approximately 20% sulfate groups. Agar is a mixture of these two polysaccharides, agarose and agaropectin [2].

Agarases are enzymes capable of hydrolyzing agar derived from sources such as seawater, marine sediments, and seaweeds. This enzyme can hydrolyze or soften agar, leading to phenomena such as surface depression or the formation of clear zones. As shown in Figure 1, agarases can be classified based on their action on various glycosidic bonds. α-agarase (EC 3.2.1.158) can hydrolyze α-(1→3) linkages to produce agaro-oligosaccharides (AOS) [3]. β-agarase (EC 3.2.1.81) can hydrolyze β-(1→4) linkages to produce neoagaro-oligosaccharides (NAOS) with β-d-galactose at the reducing end [4,5,6]. The hydrolysis products of these enzymes are primarily 3,6-anhydro-l-galactose and d-galactose [2].

*Rheinheimera* sp. encompasses a diverse group of Gram-negative bacteria found in various environments, such as seawater, brackish water, and sediments. These bacteria exhibit various characteristics, such as being aerobic, rod-shaped, and motile, with varying optimal growth conditions in terms of temperature, pH, and salinity [7,8,9]. They are known for their biofilm-forming abilities, which can affect the working of devices such as reverse osmosis modules, making them a focus of research on biofouling and anti-fouling strategies [10]. Additionally, some *Rheinheimera* strains have been identified as multidrug-resistant, containing genomic islands with antibiotic resistance genes and arsenic tolerance operons, highlighting their potential role in environmental antibiotic and arsenic pollution, and the spread of resistance genes [11]. Furthermore, *Rheinheimera* sp. has attracted attention because of its various useful enzymes. However, no studies have reported on agarases derived from *Rheinheimera* sp.

Polysaccharide mixtures containing AOS demonstrate stability due to their interaction with proteins owing to their anionic characteristics and the three-dimensional network structure of polysaccharides [12]. Despite being poorly digestible oligosaccharides, recent studies have demonstrated their potential contribution to human health. These include anticancer properties [13], antitumor effects [14], skin whitening [15], anti-inflammatory effects [15], anti-obesity effects [16], anti-fatigue properties [17], hepatoprotective effects [18], antioxidant activity [18], prebiotic function [19], and the attenuation of smooth muscle cell contraction [20]. These activities are associated with the chemical structure, molecular weight, degree of polymerization, and glycosidic bond positions of the oligosaccharides [21]. Among the studies reported thus far, only one, conducted by Shirai et al., has examined the effects of AOS on muscle recovery. That study demonstrated that AOS prevent myotube shrinkage induced by tumor necrosis factor (TNF)-α, a pro-catabolic factor, by maintaining myosin heavy-chain (MyHC) protein levels and normalizing myostatin mRNA expression [20]. Furthermore, no study has explored the effects of AOS produced by enzymes derived from *Rheinheimera*, and no study has investigated the effects of AOS produced by enzymes derived from *Rheinheimera* on muscle function. In this study, we examine the effects on muscle function of oligosaccharides produced by newly identified marine bacteria.

## 2. Results

### 2.1. The Hydrolytic Ability of Rheinheimera sp. (HY) on Polysaccharides

This study initially aimed to identify agar-degrading marine bacteria in Shimabara, Nagasaki, Japan. Through a screening of crude extracts, we found that agarases produced by marine bacteria demonstrated the highest activity at various pH values, even in the presence of metal ions.

The DNA of the selected agarase-producing hydrolytic strains was extracted by high-temperature cell lysis and used as a template. The 16S rRNA conserved sequence primer 16S-F/16S-R was used to amplify the 16S rRNA fragment. The obtained sequences were analyzed using the Basic Local Alignment Search Tool (BLAST) program in the National Center for Biotechnology Information (NCBI) database. The analysis revealed that the selected strains showed similarities of 93.68% to *Rheinheimera* sp. WMF-1 (accession no. AM690025.1). For subsequent experiments, *Rheinheimera* sp. WMF-1 will be renamed *Rheinheimera* sp. (HY). After picking colonies with a strong hydrolysis ability (obvious depressions) and conducting a streak culture to obtain single pure colonies (Figure 2A,B), it was found that *Rheinheimera* sp. (HY) exhibited hydrolytic activity on agar, agarose, and starch (Figure 2C).

### 2.2. Biochemical Characterization of Crude Enzyme

#### 2.2.1. Substrate Specificity of Crude Enzyme

Three polysaccharides, agar, soluble starch, and agarose, were chosen as the reaction substrates. The 3,5-dinitrosalicylic acid (DNS) method was used to measure the reducing sugar content and determine enzyme activity, investigating the substrate specificity of the crude enzyme (Table 1). The substrate with the highest activity was assigned a relative activity of 100%, and the activities of other substrates were expressed as relative percentages. The results showed that the crude enzyme had the highest catalytic activity against agarose, followed by agar and starch.

#### 2.2.2. Optimal Temperature and pH of Crude Enzyme Activity

In this experiment, agarose, the substrate with the highest relative activity, was used as the reaction substrate. The highest activity measured at each temperature and pH was set to 100%, and the agarase activities measured at other temperatures were expressed as relative percentages. The results showed that the catalytic activity of the crude enzyme increased with the increasing temperature, with an optimal reaction temperature of 50 °C. When the temperature was raised to 60 °C, the relative activity decreased to 53.7% (Figure 3A). The optimal pH for crude enzyme activity was 10. Activity remained relatively high between pH 6.0 and pH 11.0, retaining more than 95% of its residual activity and exhibiting considerable pH stability across a wide range of values. A statistical analysis revealed no significant differences in the enzyme activity across pH 6, 7, 8, 9, 10, and 11, with peak activity at pH 6.0. Additionally, using the pH 6.0 buffer, significant differences in the enzyme activity were observed when comparing pH 3, 4, and 5, which showed lower stability. In contrast, the enzyme remained stable and active across the broader pH range of 6 to 11. At pH = 3.0, the relative activity decreased to 65.8% (Figure 3B).

#### 2.2.3. Effects of Chemicals on Crude Enzyme Activity

Agarose was used as the substrate in this experiment. The experimental results showed that manganese sulfate (MnSO_4_) and ferrous sulfate (FeSO_4_) significantly increased the activity of the crude enzyme by 184.98% and 580.73%, respectively, while the addition of potassium chloride (KCl), sodium chloride (NaCl), urea, sodium dodecyl sulfate (SDS), and ethylenediaminetetraacetic acid (EDTA) resulted in a decrease in agarase activity (Table 2). EDTA, a metal chelator, inhibits metalloproteases. Both high and low concentrations of EDTA inhibited the activity of the crude enzyme, suggesting that the enzyme studied in this experiment is a metalloenzyme. Treatment with 5 mM SDS significantly inhibited agarase activity, likely due to the disruption of the enzyme’s hydrophobic bonds. The addition of the urea denaturant interacts with the cell wall and cell membrane, and the addition of the curing enzyme reduces the permeability of certain proteins inside and outside the cell, as well as the activity of proteases on the cell membrane. Additionally, urea might disrupt the enzyme’s hydrophilic bonds, resulting in reduced activity.

#### 2.2.4. Analysis of Hydrolytic Activity of Agarase in Crude Enzymes

This study investigated the hydrolytic activity of crude enzymes on different glycosidic bonds. Two artificial substrates, *p*-nitrophenyl-α-d-galactopyranoside (which is cleaved by enzymes that hydrolyze α-glycosidic bonds) and *p*-nitrophenyl-β-d-galactopyranoside (which is cleaved by enzymes that hydrolyze β-glycosidic bonds), were used for activity measurements. The results showed an increase in agarase activity towards α-form and β-form glycosidic bonds (Table 3), suggesting a higher activity specifically towards α-form glycosidic bonds.

### 2.3. Agaro-Oligosaccharides (AOS) on Mouse C2C12 Myotubes

#### 2.3.1. AOS on C2C12 Cell Viability

Figure 4 shows the effects of AOS on viability, using mixtures of AOS with differing molecular weights (<10 kDa, 10–30 kDa, and >30 kDa). The mixture was treated at a concentration of 2000 µg/mL. When AOS was added for 24 h, the 10–30 kDa fraction-treated C2C12 differentiated myotubes exhibited a 53% increase in viability, as indicated by the absorbance at 450 nm, compared to the control group, which was set at 100%.

#### 2.3.2. Effect of AOS on Diameters of Myotubes

To analyze the effects of AOS on the diameters of C2C12 myotubes, 10 images per group of myotube cultures were taken, using a phase-contrast microscope at 20× magnification. Myotube diameters were measured for a total of 20 myotubes from 10 fields using the BZ-II Analyzer software (BZ-7228 Keyence, Osaka, Japan), selecting the thickest portion of each myotube for measurement. The oligosaccharide mixture was treated at a concentration of 2000 µg/mL, and 2% horse serum (HS)/Dulbecco’s modified Eagle medium (DMEM) was used as the controls. Figure 5A shows microscopic images of the myotubes subjected to each treatment (AOS mixture treatment (2000 µg/mL) <10 kDa; 10–30 kDa; >30 kDa; 2% HS/DMEM medium-only control). The average diameter of the myotubes without any additives (2% HS/DMEM control) was 16.7 μm. When AOS > 30 kDa was added, the diameter was about 20.6 μm. The addition of AOS of 10–30 kDa resulted in a diameter of approximately 24.5 μm, while AOS < 10 kDa resulted in a diameter of about 20.6 μm. Compared to the control, the addition of 10–30 kDa AOS significantly increased the myotube diameter (Figure 5B).

#### 2.3.3. Relative mRNA Expression of the Genes Related to Muscle Synthesis

The MyHC composition plays a crucial role in muscle hypertrophy. Myosin is a crucial component of force-generating units in eukaryotic cells. Through ATPase activity, myosin undergoes conformational changes and binds to actin filaments, thereby producing a driving force for muscle contraction and cell movement. Myosin consists of the products of three types of genes: myosin heavy-chain, myosin light-chain, and myosin regulatory light-chain. Among these, myh3, myh4, and myh7 are predominantly expressed in skeletal muscles, whereas the non-muscle myosin genes, myh9, are located as single genes on chromosomes 5 and 11. myh3 and myh7 are expressed during mouse and human embryonic development, respectively. While myh3 expression decreases postnatally, myh7 expression persists in adult muscle. In adult skeletal muscles, myh7 is predominantly expressed in slow muscle fibers, whereas myh4 is expressed in fast muscle fibers. myh3 is not expressed in adult muscle under normal conditions, but is activated during muscle regeneration. Consequently, the expression profile of myh genes and their encoded proteins serves as a signature for specific cell types, fiber types, and the regenerative status of skeletal muscles [22].

Figure 6 shows the effect of AOS on the mRNA expression of muscle molecular motors in C2C12 myotubes. The expression levels were compared to the control group, which consisted of C2C12 myotubes cultured without AOS. Compared to the control group without AOS, the 10–30 kDa AOS significantly increased the mRNA expression of the muscle molecular motors, myh3, myh4, and myh7. Although myh9 also showed an increase in mRNA expression with 10–30 kDa AOS, it was not statistically significant.

## 3. Discussion

The crude enzyme secreted by this bacterium has broad substrate specificity, exhibiting activity towards agar, agarose, and starch (Figure 2). Previous studies have identified multifunctional enzymes with diverse activities, such as amylase, carrageenase, and neoagarobiose hydrolase, thereby expanding their application range [23,24,25]. For example, a novel multifunctional enzyme, Amy63, isolated from *Vibrio alginolyticus* 63, exhibits amylase, agarase, and carrageenase activity, facilitating substrate degradation across starch, carrageenan, and agar [25]. Another marine bacterium, *Halomonas meridiana*, has agarase, amylase, and xylanase hydrolase activities [26].

The characterization of *Rheinheimera* sp. (HY) revealed that its enzyme exhibits optimal activity at pH 10.0 and a temperature of 50 °C (Figure 3). Different agarase enzymes exhibit specific activities and different optimal pH and temperature ranges for activity. For instance, agarase from the *Acinetobacter* strain PS12B showed an optimal pH of 8.0 and a temperature of 40 °C, respectively [27]. AgWH50B from *Agarivorans gilvus* WH0801 displayed optimal conditions, which were pH 7.0 and 40 °C [28]. CaAga1 from *Cellulophaga algicola* DSM 14237 exhibited specific activities towards different agar sources and had an optimal pH of 7 and a temperature of 40 °C, with stability over a wide pH range [29]. *Rheinheimera* sp. (HY) adapts to a wide pH range of 6.0–10.0, maintaining more than 90% of its maximum activity throughout this range. Although the pH range is inconsistent, depending on the strains with agarase in different genera, the minimum values range widely from the maximum values, such as *Thalassomonas* sp. JAMB-A33, *Catenovulum* sp. X3, and *Vibrio* sp. AP-2 [30].

The effects of various chemical reagents on *Rheinheimera* sp. (HY) were also investigated. Notably, MnSO_4_ and FeSO_4_ enhanced the agarase activity (Table 2). The high activity observed may be attributed to the presence of MnSO_4_ and FeSO_4_ in the culture medium of *Rheinheimera* sp. (HY). MnSO_4_ has been shown to have varying effects on the crude agarase enzyme activity. Research indicates that MnSO_4_ can inhibit the activity of agarase enzymes, with results showing decreased activity when exposed to metal ions, such as manganese [31,32]. Additionally, the presence of MnSO_4_ can lead to secondary reactions that may affect the catalytic activities of enzymes, thereby affecting their overall performance [33].

According to a previous study, the addition of different concentrations of AOS mixtures (0.625, 2.5, and 10 µg/mL) did not lead to significant differences in muscle cell proliferation [19]. Furthermore, AOS does not exhibit cytotoxic effects in mammalian cells [20,34]. We found that the addition of the 10–30 kDa AOS fraction significantly increased cell viability (Figure 4), which may indicate enhanced mitochondrial activity, and also significantly increased the myotube diameter (Figure 5 and Figure 6). Additionally, the 10–30 kDa AOS fraction significantly increased the mRNA expression of the myosin heavy-chain genes myh3, myh4, and myh7 (Figure 6). The process by which oligosaccharides are taken up by muscle cells involves several steps and mechanisms. Although the exact pathway for AOS specifically might not be fully elucidated, the general process for oligosaccharide uptake and utilization can be described as follows. For example, glucose, a monosaccharide, is absorbed in the small intestine through specific transporters such as sodium–glucose-linked transporter 1 (SGLT1) and glucose transporter 2 (GLUT2). Once in the bloodstream, glucose is transported to muscle cells, where it is taken up by GLUT4 in response to insulin signaling. Inside muscle cells, glucose is metabolized through glycolysis to produce adenosine triphosphate (ATP) or stored as glycogen for future energy needs. This general framework of absorption, transport, and cellular uptake provides a basis for understanding how other oligosaccharides might be processed in the body [35,36]. In C2C12 cells, a model for muscle cell proliferation, it is important to consider how AOS may affect the glucose uptake and metabolism during the proliferating phase. Shirai et al. showed that AOS protects against myostatin overexpression and myosin heavy-chain degradation under inflammatory conditions, suggesting their potential role in regulating energy homeostasis and cell cycle pathways [20]. Future research should explore how AOS interacts with the glucose metabolism and insulin signaling in proliferating muscle cells to elucidate their effects on cell growth and differentiation.

Hong et al. showed that the treatment of Melan-a, derived from immortalized melanin synthesis cells, with neoagarooligosaccharides at concentrations up to 2000 µg/mL showed no significant changes in cell viability, with high survival rates observed [37]. Additionally, the neoagarooligosaccharide mixture at 4000 µg/mL also maintained a high cell survival rate of approximately 80% [37]. In our studies using HeLa cervical cancer cells, no changes in cell viability were observed even at a high concentration of 2500 µg/mL. In this study, although at high concentrations, we increased the concentration to 2000 µg/mL to ensure that any potential effects could be observed more clearly. This suggests that the concentration of AOS might be reduced by further purification. Further research is needed to evaluate the effects of AOS in future studies.

In the future, further staining with a nuclear marker will be required to accurately calculate the myotube diameter. The myotube analyzer can be used to analyze fixed differentiated myoblast cultures by nuclear and MyHC staining, enabling the assessment of physiological effects on muscle growth and metabolism by measuring relevant biomarkers and conducting histological analyses of muscle tissues. In addition, in vivo experiments should focus on administering oligosaccharides via oral, intravenous, or intraperitoneal routes. Their absorption, distribution, metabolism, and excretion should be carefully monitored to evaluate their bioavailability and pharmacokinetics. These studies will help elucidate the mechanisms by which AOS affect the muscle physiology and determine the most effective administration route for potential therapeutic applications.

## 4. Materials and Methods

### 4.1. Bacterial Isolation and Identification from Seawater

Seawater samples were collected from Shimabara City, Nagasaki Prefecture, Japan. The seawater was aspirated using a syringe needle and filtered through a 0.22 µm filter membrane. This filtration process was repeated three or four times. The filter membrane was then placed in a pre-sterilized liquid culture medium (NaNO_3_, K_2_HPO_4_, MgSO_4_, CaCl_2_, NaCl, FeSO_4_, MnSO_4_, and agar), incubated for 48 h at 30 °C, and shaken at 160 rpm. The cultured bacterial solution was subsequently diluted tenfold, and the diluted culture was spread onto a solid culture medium. The plates were incubated at 30 °C for 48 h. Sunken colonies were selected and purified using the streak plate method. The 16S rRNA gene fragment was amplified by a polymerase chain reaction (PCR) using primers specific to the conserved sequence of the 16S rRNA gene in prokaryotic organisms (16S-F/16S-R) for a strain identification analysis. The nucleotide primers used for PCR were as follows: 5′- ATGGTTTGATCATGGCTCAGATT-3′ for 16S-F and 5′- TCAGGTTACCTTGTTACGACTT-3′ for 16S-R. After confirming the quality of the PCR products, a DNA sequence analysis was conducted (Fasmac, Atsugi, Japan). The 16S rRNA gene sequences of the strains were compared using the BLAST program in the NCBI database.

### 4.2. Hydrolytic Ability of Bacteria

For plate-based activity assays of marine bacteria, a sterile toothpick was used for the selection of a single colony; bacteria were punctured in plates on a middle containing 0.3% agar, 0.3% agarose, and soluble starch with 1.5% agar, respectively. The plates were maintained at 30 °C for 48 h. Plates containing agar, agarose, and starch for checking multifunctional enzyme activities were stained with Lugol’s iodine solution, and the clear zone around the hole was visualized.

### 4.3. 3,5-Dinitrosalicylic Acid (DNS) Assay

The reducing sugar content was determined using 3,5-dinitrosalicylic acid (DNS), an aromatic compound that reacts with reducing sugars. Briefly, the procedure was as follows. Under continuous stirring, 1.6 g NaOH was added to 75 mL of distilled water, followed by 3,5-dinitrosalicyclic acid. To create 100 mL of DNS reagent, 3 g of sodium potassium tartrate was added, and the remaining volume was filled with distilled water [38].

### 4.4. Production of Crude Enzyme

The bacterial suspension was stored at −80 °C, and at the time of the experiment, it was rapidly thawed at room temperature until completely melted. A 0.5 mL portion of the bacterial solution–glycerol mixture was added to 15 mL of medium containing 0.1% agar, incubated at 30 °C, and shaken at 160 rpm for approximately 18 h. When the OD_600_ reached 0.8, 15 mL of the bacterial solution was transferred to 50 mL of medium containing 0.3% agar, incubated at 25 °C, and shaken at 160 rpm for 24 h. The culture solution was then centrifuged at 12,000× *g* for 10 min at 10 °C using a Nalgene centrifuge. The supernatant was concentrated and purified using 30 kDa Vivaspin (Vivaproducts, Littleton, MA, USA) at 5000 rpm for 10 min at 15 °C, and this was used as the crude enzyme for biochemical characterization.

### 4.5. Substrate Specificity of Crude Enzyme

In this experiment three polysaccharides, agar, agarose, and soluble starch, were chosen as reaction substrates. The activity was determined by using the 3,5-dinitrosalicylic acid (DNS) method. The yellow DNS reagent is reduced to a reddish-brown compound in the presence of reducing sugars. The color intensity of the reaction product is proportional to the concentration of reducing sugars, which can be measured by absorbance at 540 nm. Briefly, a 0.3% polysaccharide solution was prepared, and 400 µL of this solution was added to 200 µL of the crude enzyme. The mixture was reacted at 30 °C for 30 min to promote enzyme activity. Then, 400 µL of DNS reagent was added, and the mixture was reacted in boiling water at 100 °C for 10 min before being immediately cooled on ice. The reaction solution was placed in a 96-well plate, and the absorbance at 540 nm was measured using a microplate reader (BioTek Cytation 3, Winooski, VT, USA). Relative activity was calculated by setting the highest activity among the different substrates to 100%, and the activities of the other substrates were expressed as relative percentages.

### 4.6. Optimal Temperature of Crude Enzyme

In order to identify the optimal temperature for crude enzyme activity, 20 mM Na_2_HPO_4_/citric acid buffer at pH 6.0 was incubated at various temperatures (10 °C, 20 °C, 30 °C, 40 °C, 50 °C, 60 °C, and 70 °C) for 5 min. Subsequently, 200 µL of crude enzyme and 400 µL of the substrate solution were added. The reaction was carried out at the specified temperatures for 30 min. After the reaction, 400 µL of DNS reagent was added, and the mixture was heated in boiling water at 100 °C for 10 min and then immediately cooled on ice. The reaction solution was placed in a 96-well plate, and the absorbance at 540 nm was measured using a microplate reader (BioTek Cytation 3, Winooski, VT, USA). The highest activity measured at different reaction temperatures was set to 100%, and the activities at other temperatures were expressed as relative percentages.

### 4.7. Optimal pH of Crude Enzyme

To determine the optimal pH for crude enzyme activity, various buffers were prepared: 20 mM Na_2_HPO_4_/citric acid buffer (pH 3.0–5.0), 20 mM Na_2_HPO_4_/NaH_2_PO_4_ buffer (pH 6.0–7.0), 20 mM Tris/HCl buffer (pH 8.0–9.0), and 20 mM glycine/NaOH buffer (pH 10.0–11.0). Each buffer (400 µL) was mixed with 200 µL of the crude enzyme and 400 µL of the substrate solution (0.3% agar solution) in a 2.0 mL Eppendorf tube. The reactions were conducted at 30 °C for 30 min. After the reaction, 400 µL of DNS reagent was added, and the mixture was heated in boiling water at 100 °C for 10 min and then immediately cooled on ice. The reaction solution was placed in a 96-well plate, and the absorbance at 540 nm was measured using a microplate reader (BioTek Cytation 3, Winooski, VT, USA). The highest activity measured at different pH values was set to 100%, and the activities at other pH levels were expressed as relative percentages.

### 4.8. Effect of Chemicals on the Crude Enzyme

To determine the effects of various additives on the activity of the crude enzyme, 5 mM solutions of KCl, MnSO_4_, FeSO_4_, NaCl, urea, SDS, and EDTA were prepared using 20 mM Na_2_HPO_4_/citric acid buffer (pH 6.0). In a 2.0 mL Eppendorf tube, 400 µL of the 5 mM additive solution, 200 µL of the crude enzyme, and 400 µL of the substrate solution were combined. The reaction mixture was prepared to ensure that the final concentration of each additive was 5 mM throughout the reaction. The reaction was conducted at 30 °C for 30 min. Subsequently, 400 µL of DNS reagent was added, and the mixture was heated in boiling water at 100 °C for 10 min and then immediately cooled on ice. The reaction solution was placed in a 96-well plate, and the absorbance at 540 nm was measured using a microplate reader (BioTek Cytation 3, Winooski, VT, USA). The highest activity measured without additives was set to 100%, and the activities with additives were expressed as relative percentages.

### 4.9. Glycosidic Bonds of Crude Enzyme Activity

Agarases are classified into α-agarases and β-agarases based on the type of glycosidic bond they hydrolyze. To determine enzyme specificity, artificial chromogenic substrates such as *p*-nitrophenyl-α-d-galactopyranoside and *p*-nitrophenyl-β-d-galactopyranoside were used. *p*-Nitrophenol is a yellow compound that absorbs light at 420 nm. The absorbance at this wavelength is directly proportional to the amount of *p*-nitrophenol released during the reaction. Briefly, 200 µL of crude enzyme and 400 µL of the artificial substrate were incubated at 30 °C for 2 h. The reaction was terminated by the addition of 400 µL of 1 M Na_2_CO_3_. A sample of the reaction solution was transferred to a 96-well plate, and the absorbance at 420 nm was measured using a microplate reader (BioTek Cytation 3, Winooski, VT, USA). The specific glycosidic bond hydrolyzed by the enzyme was determined based on the resulting absorbance.

### 4.10. Preparation of Agaro-Oligosaccharide

Stock cultures were preserved and added to 0.1% agar medium. The cultures were incubated at 30 °C and shaken at 160 rpm for 24 h. The 0.1% agar medium cultures were then transferred to a 0.3% agar medium and incubated under the same conditions (160 rpm, 30 °C) for 48 h. The cultures were centrifuged at 12,000× *g* for 10 min at 10 °C to separate the supernatant from the cell debris. The supernatant was concentrated using a Vivaspin centrifugal concentrator at 5000× *g* for 10 min at 15 °C. The concentrate was fractionated into three molecular weight ranges: <10 kDa, 10–30 kDa, and >30 kDa. The fractionated samples were then freeze-dried. The initial freeze-drying was conducted for 24 h, followed by a secondary freeze-drying for 48 h. Freeze-dried samples were dissolved in Milli-Q water. The oligosaccharide solutions so dissolved were filtered through a 0.22 μm filter to remove any remaining particulate matter. The filtered oligosaccharide solutions were stored at −80 °C.

### 4.11. Cell Culture

The C2C12 myoblast cells used in this study were purchased from the American Type Culture Collection (Rockville, MD, USA). C2C12 is a myoblast cell line derived from mice. They were cultured in DMEM containing 10% fetal bovine serum (FBS) and 1% penicillin–streptomycin and incubated at 37 °C with 5% CO_2_. When the C2C12 myoblasts reached 100% confluence, the medium was changed to DMEM supplemented with 2% horse serum (HS). Cells were maintained in 2% HS/DMEM (differentiation medium) for three days for the formation of myotubes.

### 4.12. Cell Viability Assay

Cell viability was measured using the cell counting kit-8 (CCK-8) (Dojindo, Kumamoto, Japan). The CCK-8 assay measures cell viability based on the activity of cellular dehydrogenase enzymes. These enzymes reduce the water-soluble tetrazolium salt, WST-8, into a yellow-colored formazan product. The amount of formazan produced is directly proportional to the number of metabolically active (viable) cells in the sample. C2C12 myotubes were treated with AOS (final concentration: 2000 μg/mL) for 24 h. The culture medium was then removed, and the cells were washed three times with Hank’s balanced salt solution (HBSS) (Gibco, Grand Island, NY, USA). Following treatment with CCK-8 (10 μL/100 μL HBSS) in each well for 30 min, absorbance at the wavelength of OD_450_ nm was read using a microplate reader (BioTek Cytation 3, Winooski, VT, USA) and imager software (Version 6.1).

### 4.13. Measurement of Myotube Diameter

C2C12 myoblasts were plated in 24-well plates (5 × 10^4^ cells/well) and then differentiated myotubes were treated with AOS (freeze-dried samples were dissolved in Milli-Q water (Millipore, Burlington, MA, USA)) at a final concentration of 2000 mg/mL for 24 h, after which they were photographed at ×20 magnification. The myotube diameters were of 20 randomly measured myotubes from 10 fields using the BZ Analyzer (Keyence, Osaka, Japan), selecting the thickest portion of each myotube for measurement. The oligosaccharide mixture was treated at a concentration of 2000 µg/mL, with 2% HS/DMEM for the control for C2C12. The BZ-H3C/Hybrid Cell Count option was used to measure the diameter of the myotubes. The exposure time was set to 1/50 s and images were acquired at a bit depth of 12 bit, black-and-white, using a BIOREVO BZ-X710 microscope (Keyence, Osaka, Japan).

### 4.14. Quantitative Real-Time Polymerase Chain Reaction (qRT-PCR)

The total RNA was extracted from the cells using an acid guanidinium thiocyanate-phenol–chloroform mixture. A qRT-PCR analysis was performed with the appropriate primers and SYBR Green dye using a real-time PCR system (ABI Real-Time PCR Detection System; Applied Biosystems, Foster City, CA, USA) [39]. The mRNA levels were normalized to the housekeeping gene 18S ribosomal RNA. The oligonucleotide primers used for PCR were as follows: 5′- CCAACAGACTCCTGGCACAT-3′ and 5′- CTGAACAGTGCAGAGACGGT-3′ for myh3 (NM_001099635); 5′- GGAGTTCACACGCCTCAAAGAG-3′ and 5′- TCCTCAGCATCTGCCAGGTTGT-3′ for myh7 (NM_001361607); 5′- ATCCTGGAGGACCAGAACTGCA-3′ and 5′- GGCGAGGCTCTTAGATTTCTCC-3′ for myh9 (NM_022410); 5′- GACAGCCAAGAAGAGGAAACTGG-3′ and 5′- ACCTGCCATCTCTTCTGTGAGG-3′ for myh4 (NM_010855); and 5′- GTAACCCGTTGAACCCCATT-3′ and 5′- CCATCCAATCGGTAGTAGCG-3′ for mouse 18S. The qPCR parameters were set as follows: 50 °C for 2 min, 95 °C for 10 min, followed by 40 cycles of 95 °C for 15 s, and 60 °C for 1 min. The final melt curve analysis was conducted at 95 °C for 15 s, 60 °C for 1 min, and 95 °C for 15 s. Calculations for qPCR experiments were performed using absolute quantification.

### 4.15. Statistical Analysis

All data are represented as mean ± S.D. and were analyzed by a one-way or two-way analysis of variance (ANOVA) using IBM SPSS statistics version 26.0 software (Armonk, NY, USA), followed by Tukey’s test for individual differences between groups. Statistical significance was set at *p* < 0.05.

## 5. Conclusions

In this study we screened and identified agarase-producing bacterial strains in seawater samples, including *Rheinheimera* sp. (HY), that showed significant agarolytic activity. Optimal culture conditions for the strain were determined, and the best growth was observed at 30 °C on 0.1% agar. The biochemical characterization of the crude enzyme revealed the highest catalytic activity for agarose, with an optimal reaction temperature of 50 °C and a pH of 10.0. MnSO_4_ and FeSO_4_ notably enhanced enzyme activity, whereas other additives, such as KCl, NaCl, urea, SDS, and EDTA, reduced it. The enzyme demonstrated broad substrate specificity, hydrolyzed agar, agarose, and starch, and effectively broke down α-glycosidic and β-glycosidic bonds. Furthermore, this study explored the effects of AOS on mouse C2C12 myotubes, revealing that the 10–30 kDa agaro-oligosaccharide fraction significantly increased cell viability, the myotube diameter, and the expression of myosin heavy chains. These findings suggest that AOS can enhance the muscle fiber diameter following muscle differentiation. To better assess the myotube formation and the proliferation of myoblasts, future studies should incorporate a nuclear marker to calculate the fusion index, which would provide a more accurate measurement of myoblasts’ ability to differentiate into myotubes. These studies will help elucidate the mechanisms by which AOS affect the muscle physiology and determine the most effective administration route for potential therapeutic applications.

## Figures and Tables

**Figure 1 marinedrugs-22-00515-f001:**
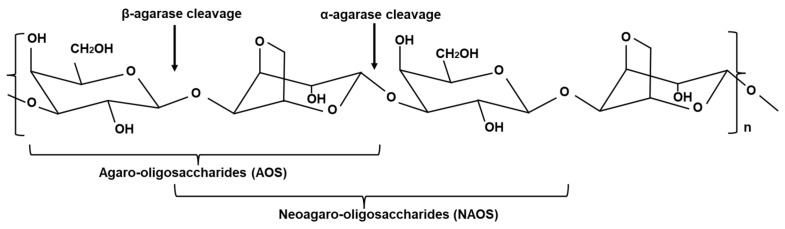
Structure of agar (Chen et al., 2021 and Xu et al., 2018, partly modified) [5,6].

**Figure 2 marinedrugs-22-00515-f002:**
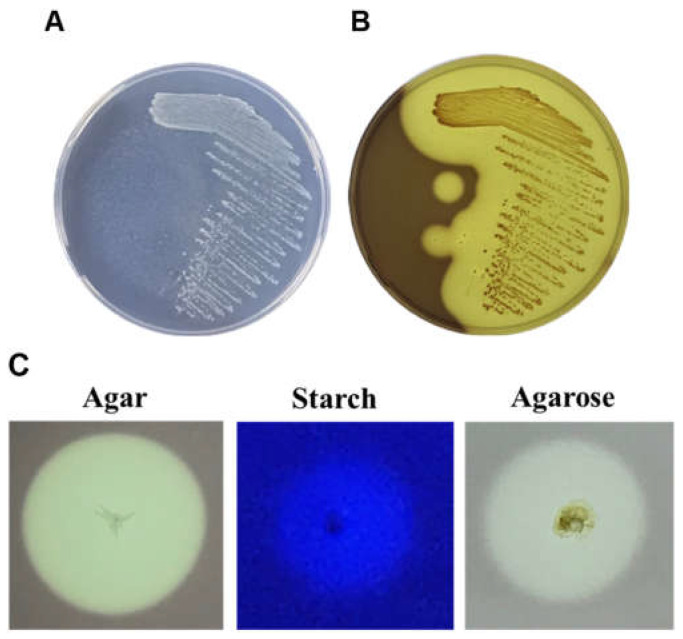
Agarolytic activity of *Rheinheimera* sp. (HY) isolated from seawater. (**A**) Marine bacterium *Rheinheimera* sp. (HY) on an agar plate, incubated for 48 h. (**B**) The hydrolysis zone was colored with Lugol’s iodine solution after static incubation for 48 h on an agar plate. (**C**) Plate-based hydrolytic activity assays for agar, starch, and agarose.

**Figure 3 marinedrugs-22-00515-f003:**
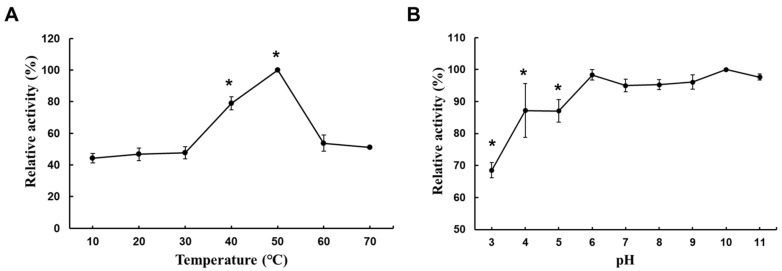
Optimal temperature and pH for crude enzyme activity. (**A**) Effect of temperature on agarase activity. (**B**) The optimal pH of agarase’s crude enzyme was measured in 20 mM Na_2_HPO_4_/citric acid (pH 3.0, 4.0, and 5.0), 20 mM Na_2_HPO_4_/NaH_2_PO_4_ (pH 6.0, 7.0, and 8.0), 20 mM Tris/HCl (pH 9.0), and 20 mM Glycin/NaOH (pH 10.0 and 11.0). The mixture was reacted at 30 °C for 30 min to promote enzyme activity by the DNS method. Data are expressed as the mean of triplicate measurements with standard deviation (*n* = 3, mean ± SD). Data are expressed as mean ± SD. Significance: (**A**) * *p* < 0.01, compared with 30 °C. (**B**) * *p* < 0.01, compared with pH = 6.

**Figure 4 marinedrugs-22-00515-f004:**
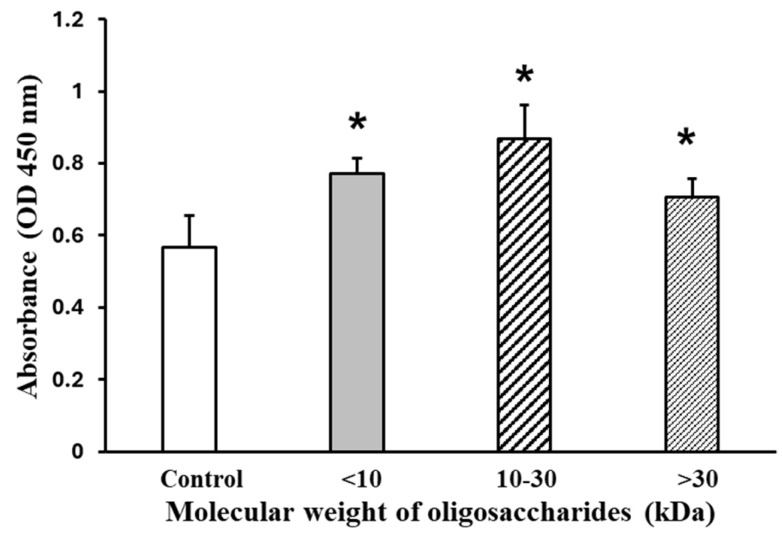
AOS on C2C12 myotube viability. The myotubes were treated by the AOS mixtures at a concentration of 2000 µg/mL for 24 h. Cell viability was quantified using the Cell Counting Kit-8 (CCK-8) assay. Data are represented as mean ± S.D. (*n* = 8). Significance: * *p* < 0.01, compared with control (2% HS/DMEM only culture medium) group.

**Figure 5 marinedrugs-22-00515-f005:**
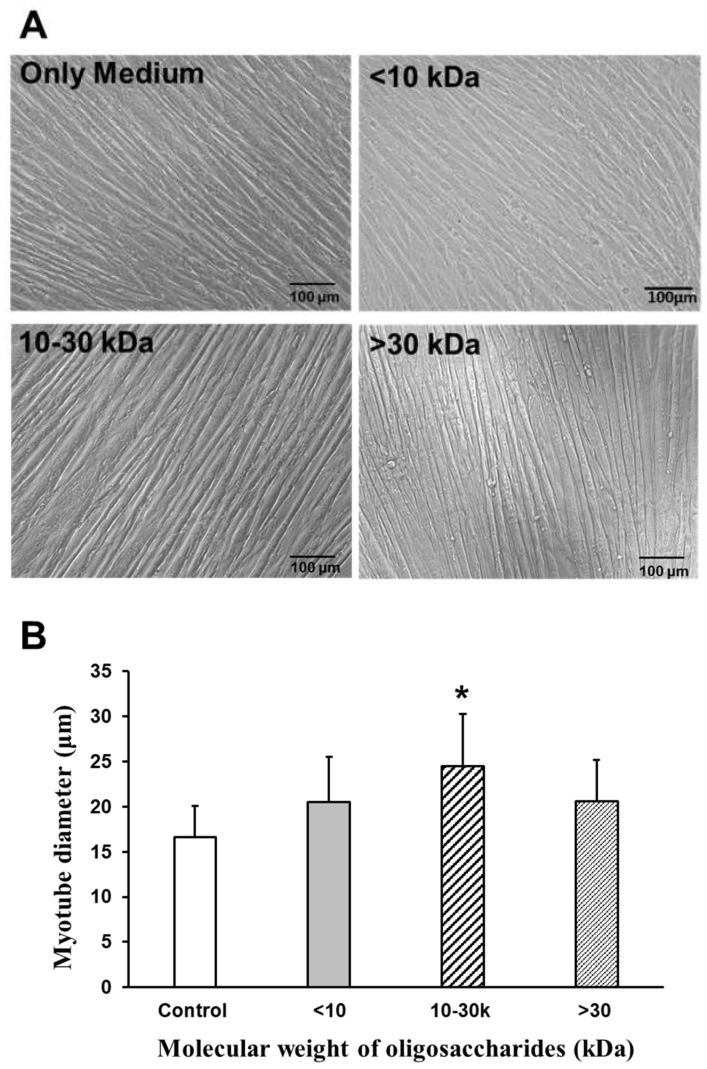
Effect of AOS on the diameters of myotubes. (**A**) Diameters of myotubes in the absence or presence of oligosaccharide mixture for 24 h. Images of myotubes at ×20 magnification; <10 kDa, 10–30 kDa, and >30 kDa oligosaccharide AOS mixture used (at 2000 µg/mL), and 2% HS/DMEM for control. Scale bars indicate 100 µm. (**B**) Diameters of myotubes in the absence or presence of the oligosaccharide mixture. Myotube diameters were 20 randomly measured myotubes from 10 fields with BZ Analyzer (Keyence, Osaka, Japan). C2C12 myotubes were treated with the AOS mixtures at a concentration of 2000 µg/mL or no treatment as control. Data are represented as mean ± S.D. Significance: * *p* < 0.01, compared with control group (2% HS/DMEM-only culture medium).

**Figure 6 marinedrugs-22-00515-f006:**
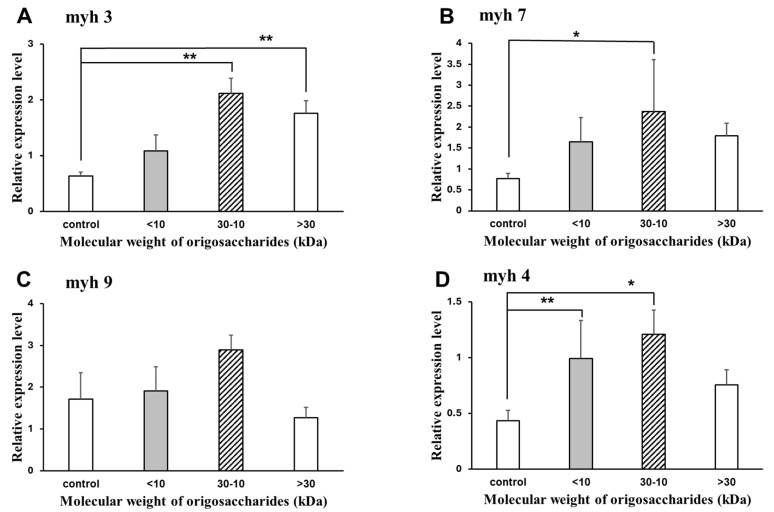
Effect of agaro-oligosaccharide mixture on the expression of myosin heavy-chain genes. Myotubes were incubated with the indicated concentrations of 2000 µg/mL oligosaccharides for 24 h. The ratio between the intensities of the myosin heavy-chain genes and 18S rRNA was calculated. Relative myh3 (**A**), myh7 (**B**), myh9 (**C**), and myh4 (**D**) mRNA were determined by real-time reverse transcription-polymerase chain reactions. Data are represented as mean ± S.D. (*n* = 4). Significance: * *p* < 0.05 and ** *p* < 0.01, compared with control group (2% HS/DMEM-only culture medium).

**Table 1 marinedrugs-22-00515-t001:** Substrate (agar, agarose, and starch) specificity of *Rheinheimera* sp. (HY) agarase crude enzymes.

Substrate	Relative Activity (%) ^a,b,c^
Agarose	100.00 ± 0.01
Agar	68.51 ± 2.50 ^d^
Starch	28.85 ± 3.58 ^d^

^a^ The mixture was reacted at 30 °C for 30 min. The activity of agarase was then measured according to the amount of reducing sugar by DNS method. ^b^ The highest enzymatic activity was defined as 100%. ^c^ All data mean values from triplicate experiments. ^d^ Data are represented as mean ± S.D. (*n* = 3). *p* < 0.05, compared with Agarose.

**Table 2 marinedrugs-22-00515-t002:** The effect of chemicals on *Rheinheimera* sp. (HY) agarase’s crude enzyme activity.

Reagents	Relative Activity (%) ^a,b,c^
Concentration (5 mM)	
None	100.00 ± 0.01
KCl	22.59 ± 4.50 ^d^
MnSO_4_	184.98 ± 12.55 ^e^
FeSO_4_	580.73 ± 144.45 ^e^
NaCl	54.63 ± 9.07
EDTA	30.13 ± 4.12 ^d^
SDS	77.34 ± 5.70
Urea	51.86 ± 8.51

^a^ The mixture was reacted at 30 °C for 30 min. The activity of agarase was then measured according to the amount of reducing sugar by DNS method. ^b^ The highest enzymatic activity was defined as 100%. ^c^ All data mean values from triplicate experiments. ^d^ Data are represented as mean ± S.D. (*n* = 3). *p* < 0.05, compared with None group. ^e^ Data are represented as mean ± S.D. (*n* = 3). *p* < 0.01, compared with None group.

**Table 3 marinedrugs-22-00515-t003:** Glycosidic bond analysis of *Rheinheimera* sp. (HY) agarase’s crude enzyme activity using artificial chromogenic substrates.

Artificial Chromogenic Substrates	Absorbance (OD 420 nm) ^a^
	48 h
*p*-nitrophenyl-α-d-galactopyranoside	0.095 ± 0.010
*p*-nitrophenyl-β-d-galactopyranoside	0.071 ± 0.014

^a^ The mixture was reacted at 30 °C for 2 h and terminated by the addition of 1 M Na_2_CO_3_. Crude enzymes were from 48 h incubation.

## Data Availability

The data presented in this study are available on request from the corresponding author.

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
