# Peer review of "The Effects of Agaro-Oligosaccharides Produced by Marine Bacteria (*Rheinheimera* sp. (HY)) Possessing Agarose-Degrading Enzymes on Myotube Function"

_marinedrugs, 2024, doi:10.3390/md22110515_

Round 1

Reviewer 1 Report

Comments and Suggestions for Authors

The paper describes the effect of AOS produced by marine bacteria. Specifically of interest is their impact on myotubes. The findings are of interest, the paper is generally well-written. The comments provided are meant to increase the effect on a broader audience and clarify the message.

Line 23: The myotubes do not exhibit proliferation. They are the final differentiated form of muscle cells thus they do not divide nor proliferate. They can, however, change their size and become hypo or hypertrophic. Please modify the sentence accordingly.  

32: is starch not one of the main polysaccharides also?

63: how can AOS show excellent “protein stability”?? Are they not polysaccharides?

111: Table 1: Please provide the error type and number of repetitions.

122: based on the data I would not say “below pH 3” but exactly “at pH = 3”

144: Table 2: Please provide the error type and number of repetitions.

145: basic principles/mechanism of what 420nm absorbance means should be given so that data can be understandable to a broader audience

147-149: unclear sentence: how can substrate hydrolyzes the bonds….please rewrite.

150: based on data it seems some better activity towards alpha bonds.

151: Table 3: Please provide the error type and number of repetitions. The absorbance values seem rather low. Did you check if the instrument has appropriate sensitivity to reliably measure such low absorbance values?

154: basic principles/mechanism of what 450nm absorbance means should be given so that data can be understandable to a broader audience

155: the myotubes do not divide i.e. proliferate thus the cell number (nuclei) is constant. Would be, however, interesting if the authors would treat the C2C12 cells in the myoblast stage to observe the effect on cell proliferation. I encourage the authors to perform such an additional experiment as it would benefit the paper and the final message significantly.

157-158: unclear sentence, please rewrite. What did increase by 53%?

162: Figure 4: Please provide the error type and number of repetitions.

163: unclear sentence. The myotubes were treated by the AOS mixtures at a concentration of 2000µg/ml. ?

168: Would be interesting to know how the myotubes were “randomly selected”?

170: was this a semi- or full-automatic analysis? More details are needed.

185: Figure 5: how can one achieve to culture 20 myotubes at density 10cells/well? Very unclear sentence, please rewrite it.

186: control: the control was (probably) complete differentiation media i.e. DMEM + 2 % horse serum where MilliQ water was added to simulate the addition of AOS? Unclear sentence please rewrite.

205/207: “Growth factors” is not the best word to describe myosins. Please use more appropriate expressions, like molecular motors, etc.

211: It is not clear to what was expression compared. Relative to what? Please provide more details! Listed genes are not “myosin-related” but they ARE encoding myosin [heavy chain]. Please adjust accordingly.

217: DMEM only culture medium: was not differentiation medium used as a control?

255: Listed genes are not “myosin-related” but they ARE encoding myosin [heavy chain]. Please adjust accordingly.

258-266: The paragraph describes the possible uptake of AOS in the body. However, the authors should provide some details of possible AOS uptake by C2C12 myotubes also!

285: primers should be listed

304: unclear sentence, please rewrite. How can ice melt at -80C?

320: The basic principle of absorbance measurement at 540 nm should be stated!

236-327: How was the T of the crude enzyme and substrate controlled before adding to the buffer of the desired T? Was some T equilibration considered?

347: not clear if 5mM was the final concentration (after the addition of crude enzyme and substrate) or the initial concentration which was diluted by crude enzyme and substrate addition.

364: the basic principle of absorbance measurement at 420 nm should be stated.

396: how can one achieve culturing 20 myotubes at a density 10 wells/well? Very unclear sentence, please rewrite it.

397-398: another unclear sentence. Please rewrite it.

398: Would be great to see how the image processing was done, perhaps some depiction in one image?

400: Please state exposure time, bit-depth, brightfield techniques…

405: Ref needed (after “described previously”).

408-412: please provide accession codes for the myosin genes also.

432: myosin related genes -> myosin heavy chains

434-435: how can a nuclear marker help more accurately determine diameter? What I think would be good to do is to determine the fusion index – the ability of myoblast to form myotubes. And the proliferation of myoblasts themselves

Comments on the Quality of English Language

It is generally good, but certain sentences are unclear and need to be corrected.

Author Response

We are attaching the Point-by-point response to the comments of Reviewer 1.

Reviewer 2 Report

Comments and Suggestions for Authors

This manuscript describes the effects of agaro-oligosaccharides produced by marine bacteria (Rheinheimera sp.) in C2C12 myotubes. The study present new data about the action of an oligosaccharides in skeletal muscle cells. The paper is well written however I have some major and minor comments what should be addressed.

Major issues.

1.    The biggest shortcoming of the manuscript comes from the presentation of the cell culture study design. It is completely unclear what the sentences mean in row 185-186. What type of well was used in which 10 (or 20?) myotubes could be placed and cultured? Why the Authors used the MQ group (Figure 5)? Was the AOS diluted in DMEM or water? What happened in the cell culture when MQ was added? These important information is missing from the description of the study. See also row 397-398.

2.    My next problem is the inappropriate statistical analysis. The statistical comparison is missing from all tables, Figure 3 and 4. Maybe in the latter the statistics is present, but the meaning of “a, b, c, ab” is missing.

3.    I would suggest to modify the statement about pH dependence of the enzyme activity (row 120). It is clearly visible in Figure 3B that the maximum has already been reached at pH=6. If the Authors employ the statistical analysis this will be proven.

4.    I miss the explanation of the usage of the very high dose (2000 mg/ml) AOS. Previous study (reference 20) used only 10 mg/ml as the maximal dose.

5.    I suggest to use the “10-30” instead of “30-10” nomenclature. It is a very unusual way to give a range for molecular weight. This would clarify the Abstract (row 23) too.

6.    The text in row 212-217 should be moved in the legend of Figure 6.

7.    The Authors discuss the potential pathway of AOS actions in maturated skeletal muscle (row 258-266). Since the experiments were done in C2C12 cell cultures, a similar brainstorming should be included here in case of proliferating cells.

8.    It was strange to me that The Authors give details of future experiments in the Conclusion. This may fit in the Discussion, but the Conclusion should be more general.

Minor

1.    Please delete or explain the statement in row 159-160 and 252-253.

2.    Please give the number of cell cultures used in the myotube diameter experiments.

3.    Please change “treated” to “used” in row 186.

4.    Please use “myh” instead of “Myh” in Figure 6.

5.    Please give the specification of the microplate reader in row 393.

6.  Please change “ppm” to “mg/ml” in row 389.

Author Response

We are attaching the Point-by-point response to the comments of Reviewer 2

Round 2

Reviewer 2 Report

Comments and Suggestions for Authors

The manuscript has been improved however I still have some major and minor comments what should be addressed.

Major issues.

1.    Thank you for the statistical analysis. I would suggest to use stronger statistical comparison in some cases (Table 2). It is strange that there is no significant difference between control and MnSO4 and FeSO4. It is not clear what is the difference between * and # in Figure 4? Finally, I would insert the “Data are represented as mean ± S.D.” sentence in the “4.15. Statistical analysis” since this is true for all values in all table.

2.    I think a “no” is missing in the following sentence in row 129-130: “Statistical analysis revealed NO significant differences in enzyme activity across pH 6, 7, 8, 9, 10, and 11,”

3.    I still miss the explanation of the usage of the very high dose (2000 mg/ml) AOS. Previous study (reference 20) used only 10 mg/ml as the maximal dose.

4.    It is clear now why the Authors show data from the MQ group in Figure 5, but I miss the same group in Figure 4. This is a very important data which can prove that MQ alone does not modify the viability of myotubes.

Minor

1.    Please give the number of cell cultures used in the myotube diameter experiments.

2.    I would suggest to change the order of columns in Figure 4, 5, and 6 as follows: Control, <10, 10-30, >30. This is more logical then the decreasing order.

Author Response

We are attaching the Point-by-point response to the comments of Reviewer 2.
